# VISUALLY-GROUNDED HUMANOID AGENTS

**Visually-grounded Humanoid Agents in Realistic 3D Scenes**

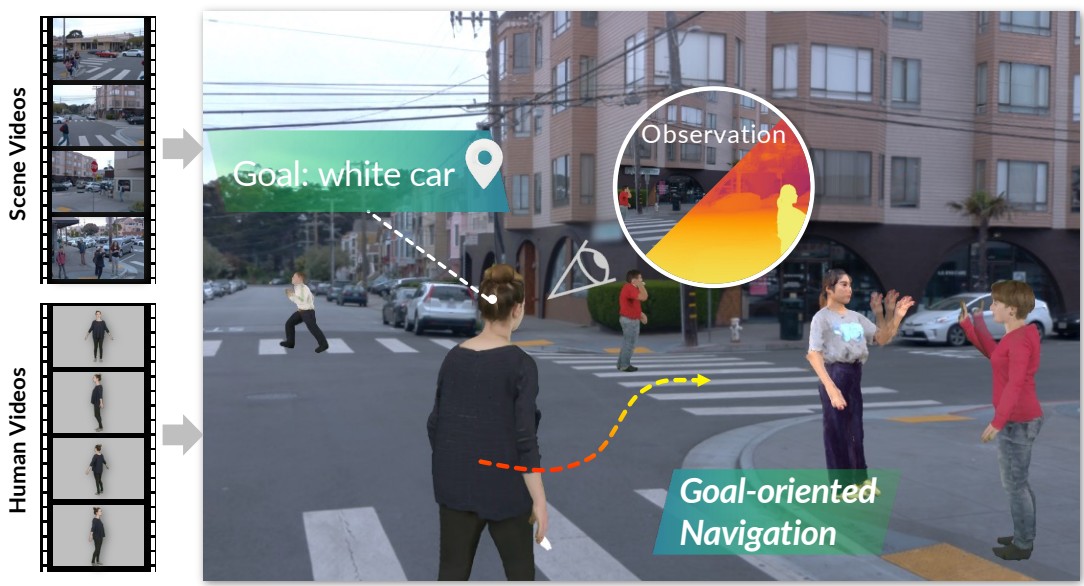

Figure 1: **Visually-grounded Virtual Humans.** This system enables humanoid agents behave actively in novel environments captured by videos.

## ABSTRACT

Digital human generation has been studied for decades and supports a wide range of real-world applications. However, most existing systems are **passively** animated, relying on privileged state or scripted control, which limits scalability to novel, unseen environments. We instead ask: how can digital humans **actively** behave using only *visual observations* and *specified goals* in novel scenes? Achieving this would enable populating any 3D environments with any digital humans, at scale, that exhibit spontaneous, natural, goal-directed behaviors. To this end, we introduce **Visually-grounded Humanoid Agents**, a coupled two-layer (world-agent) paradigm that replicates humans at multiple levels: they *look, perceive, reason, and behave* like real people in real-world 3D scenes. The World Layer provides a structured substrate for interaction, by reconstructing semantically rich 3D Gaussian scenes from real-world videos via an occlusion-aware semantic scene reconstruction pipeline, and accommodating animatable Gaussian-based human avatars within them. The Agent Layer transforms these avatars into autonomous humanoid agents, equipping them with first-person RGB-D perception and enabling them to perform accurate, embodied planning with spatial-awareness and iterative reasoning, which is then executed at the low level as full-body actions to drive their behaviors in the scene. We further introduce a benchmark to evaluate humanoid–scene interaction within reconstructed 3D environments. Experimental results demonstrate that our agents achieve robust autonomous

behavior through effective planning and action execution, yielding higher task success rates and fewer collisions compared to both ablations and state-of-the-art planning methods. This work offers a new perspective on populating scenes with digital humans in an active manner, enabling more research opportunities for the community and fostering human-centric embodied AI. Data, code, and models will be open-sourced. Project page: https://VGHuman-anonymous-ICLR26.github.io.

## 1 INTRODUCTION

Digital humans have become indispensable in AR/VR (Subramanyam et al., 2020), telepresence (Lawrence et al., 2024), and robotic training (Tsoi et al., 2020) *etc.*, where they serve as avatars for interaction and simulation. However, the majority of existing models are trained from **third-person** data (*e.g.,* motion capture systems (Cheng et al., 2023) or videos (Alldieck et al., 2018b)) that capture appearance and kinematics but neglect underlying *decision-making context*. As a result, these systems remain **passively controlled**: they replay scripted motions or follow trajectory planners *without autonomy*. Such passivity prevents digital humans from adapting to novel environments, posing a fundamental challenge for scaling up the utility of digital humans to generalize across diverse environments and use cases.

To move beyond such limitations, we argue that digital human modeling should also be framed around **active agents** that replicate humans at multiple levels: **look, perceive, move**, and **reason** like real humans in realistic 3D worlds. This requires a digital human 1) perceiving and acting from a first-person perspective using egocentric sensory input, 2) behaving adaptively under autonomous decision-making, and 3) interacting with realistic environments in a goal-directed manner. Real humans rely on their own visual observations and short-term goal positions to make context-aware choices (Warren Jr & Hannon, 1988), such as navigating cluttered sidewalks and plazas in cities. Embedding this context-aware perception–action loop is essential for developing digital humans that generalize to novel environments and exhibit purposeful behavior. While similar principles have driven progress in robotics (Anderson et al., 2018; Xie et al., 2025), the complexity of human embodiment and animation has left these questions underexplored in the digital human domain.

With large language models (LLMs) (Kaplan et al., 2020) and vision-language models (VLMs) (Radford et al., 2021), it has become straightforward to utilize these methods to simulate aspects of digital agents such as high-level reasoning (Yang et al., 2024a) and dialogue (Park et al., 2023) in complex scenes. However, such systems largely remain disembodied: they are typically constrained to symbolic reasoning (Shridhar et al., 2020) or scripted scenarios (Puig et al., 2018) and often lack visual grounding, real-world perception–action coupling, and context-aware adaptability. Some efforts integrate VLMs with visual inputs for the motion planning of agents (Cheng et al., 2024), but relying solely on VLM makes it challenging to operate effectively in complex environments. Due to these limitations, no prior work has been able to span the spectrum from semantic reasoning to embodied digital humans with perception, decision, and action fused into a continuous cycle, allowing them to adapt and act autonomously within complex, real-world 3D environments.

To this end, we introduce **Visually-grounded Humanoid Agents**, a coupled two-layer paradigm for embodied digital humans (Fig. 2) that requires only real-world source videos of scenes and humans. *1)* The base is **World Layer**, which reconstructs large-scale, semantically enriched, and compositional 3D environments from real-world captured videos and accommodates animatable human avatars within them. This layer provides the physical and semantic substrate, a realistic stage on which agents can be embodied. *2)* On top, an **Agent Layer** equips the avatars with first-person visual perception and goal-driven planning capabilities, enabling them to act autonomously and adaptively in complex 3D environments. This layer provides a perception–action loop, coupling observation, decision, and motor control into a unified cycle. The technical methods behind the two-layer paradigm address key challenges in building embodied digital humans:

1. For the scenes in World Layer, agents require environments that are simultaneously photorealistic, semantically structured, and compositional, yet existing large-scale 3D reconstructions often suffer from

occlusions and incomplete semantics. We tackle this with an **occlusion-aware semantic scene reconstruction** pipeline. It augments 3DGS (Kerbl et al., 2023) with hierarchical semantic features (Qin et al., 2024; Wu et al., 2024b) via incorporating occlusion-aware masks and view selection for feature learning. This automatic workflow boosts the accuracy of 3D instance segmentation and can segment 3D instances and annotate them with semantic descriptions in large scenes, creating a semantically rich environment.

2. For the humanoids in Agent Layer, it must couple perception, planning, and action in a unified loop, while directly relying on VLMs for decision-making lacks grounding, suffers from limited memory, and is detached from action. We overcome these limitations within the agent layer through a new **spatially-aware visual prompting** and **iterative reasoning** scheme, enabling memory-enhanced, context-aware planning that remains grounded in first-person perception. Coupled with diffusion-based motion generation for realistic full-body execution, this design closes the perception–action loop. Together with the World Layer, it yields the first paradigm for embodied digital humans that can perceive, decide, and act autonomously in complex 3D environments.

Finally, we introduce a new benchmark to evaluate how well the embodied digital humans interact with reconstructed 3D scenes. Extensive experiments demonstrate that humanoid agents can perform adaptive, reliable, and complex reasoning in 3D real-world environments, validating the effectiveness of our two-layer paradigm. We believe this work will establish the foundation for the systematic, large-scale creation and evaluation of future embodied human systems.

## 2    RELATED WORK

**Embodied Agent.** Driving embodied agents to perceive, reason and interact with the environment has been widely studied. However, most prior efforts focus on general, non-human embodiments, either by training vision–language–navigation (VLN) models or by leveraging pre-trained VLMs with carefully designed textual and visual prompts (Zhang et al., 2024a; Yang et al., 2024b; Liu et al., 2024b; Goetting et al., 2024). For example, Navid (Zhang et al., 2024a) trains a VLN model to predict next actions based on video inputs, while VLMNav (Goetting et al., 2024) uses navigability masks and potential action paths as visual prompts to guide pre-trained VLM in selecting optimal actions. However, extending these formulations for human agents is challenging due to their complex action space and diverse behavior. One alternative is to train VLN models to predict joint actions (Cheng et al., 2024), but this requires a substantial amount of paired visual–action data. Another line of work attempts to populate scenes with humans (Zhang & Tang, 2022); although these methods achieve plausible motions, they lack active scene perception and interaction, limiting their ability to adapt behaviors dynamically to the environment. Some work populates humanoid agents in scenes (Puig et al., 2021; 2023; Wu et al., 2024a). However, they are mainly made in simulation environments, where their behaviors are either scripted or based on privileged information of scenes. This makes this method hard to generalize in novel real-world environments at scale.

**3D Semantic Scene Reconstruction.** Advances in neural reconstruction like NeRF (Mildenhall et al., 2021) and 3D Gaussian Splatting (3DGS) (Kerbl et al., 2023), have achieved impressive results. Recent work has scaled 3DGS to large scenes using a divide-and-conquer strategy for scene partitioning (Kerbl et al., 2024; Ren et al., 2024; Liu et al., 2024c). Other efforts extended 3DGS to model 4D dynamic urban scenes, capturing moving objects like vehicles, pedestrians, and cyclists (Zhou et al., 2024; Lu et al., 2024; Chen et al., 2024). In parallel, progress has been made in semantic understanding of reconstructed scenes. LeRF (Kerr et al., 2023) embeds CLIP features into neural fields for text-based querying. ConceptGraph (Gu et al., 2024) integrates Grounded-SAM to construct a 3D semantic graph, while OpenGaussian (Wu et al., 2024b) uses SAM-generated masks combined with CLIP features to enable open-vocabulary scene understanding. However, they are limited to indoor scenes, while reconstructing semantically understandable outdoor scenes remains an open challenge due to vast scale and severe occlusions.

**Human Avatar Modeling.** Prior works can be categorized into three parts: appearance modeling, human animation, and human motion generation. 3DGS has been the dominant approach for realistic human appearance modeling and animation (Hu et al., 2024). For example, GaussianAvatar (Hu et al., 2024; Jiang

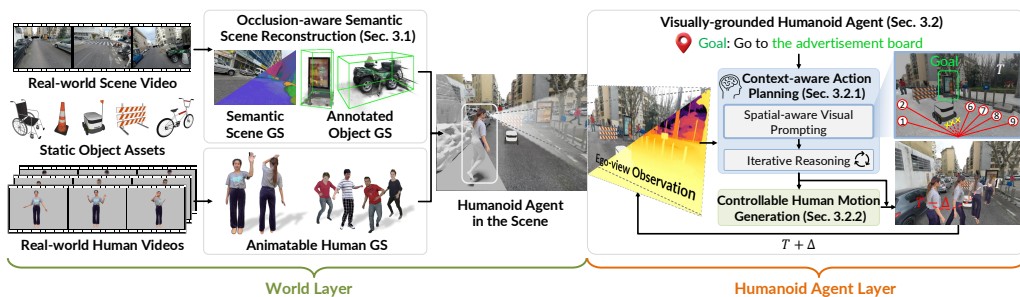

Figure 2: **Overview of our framework.** This system first creates large-scale, semantically detailed digital environments and animatable human avatars in its **World Layer**. Then, the **Agent Layer** controls these avatars through a perception-action loop for human-scene interaction.

et al., 2023; Lei et al., 2024) reconstructs animatable 3D Gaussians from monocular videos. Regarding human motion generation, recent studies largely employ diffusion models to synthesize motions from various conditional inputs (Tevet et al., 2022; Chen et al., 2023). To achieve scene-aware motion generation, some approaches incorporate scene context as a conditioning signal (Jiang et al., 2024; Hassan et al., 2021; Mir et al., 2024; Zhang et al., 2024b; Zhao et al., 2022; Cong et al., 2024); however, this typically demands large-scale datasets of human–scene interactions, which are scarce in outdoor environments.

## 3 METHOD

As illustrated in Fig. 2, the framework consists of two layers. Given input sources from the real world (scene videos, object assets, and human videos), the **World Layer** (Sec. 3.1) reconstructs: 1) large-scale, semantically enriched environments via a *occlusion-aware semantic scene reconstruction* pipeline (Sec. 3.1.1), which augments the 3DGS scenes with robust semantic annotations and spatial landmarks; and 2) *animatable Gaussian-based human avatars* (Sec. 3.1.2), which are randomly placed into the reconstructed scenes to enable scalable human–scene interaction. On top, the **Agent Layer** (Sec. 3.2) governs these avatars through a perception–action loop: First, a *high-level, context-aware planning* (Sec. 3.2.1) utilize a VLM in a zero-shot manner to interpret first-person observations through spatial visual prompting and iterative reasoning, thereby mitigating its limitations in grounding and memory, and ultimately formulating context-aware goals. Then a *low-level, diffusion-based motion generation* (Sec. 3.2.2) translates these abstract goals into realistic full-body behavior. The generated motion is executed in the environment and fed back into the agent's perception, updating observations and closing the **perception–action loop** for continuous interaction.

### 3.1 THE BASE - WORLD LAYER

We first adopt CityGaussian (Liu et al., 2024c;d) to reconstruct large-scale street scenes. To enable human–scene physical interaction with accurate collision detection, we bake it into a high-fidelity mesh (see details in the supplementary). After building a digital twin of the real-world outdoors scenes, the next step is to enrich it with semantic understanding so that humanoid agents can recognize distinct objects and landmarks, reason about their roles in context, and interact with the environment in a goal-directed manner. Directly scaling recent works (Qin et al., 2024; Shi et al., 2024; Wu et al., 2024b) from indoor scenes to complex urban environments presents two major challenges: 1) *Severe occlusion*, where objects are frequently hidden by others (e.g., a car blocking a fire hydrant), leading to incomplete geometry and unreliable semantic labeling for the occluded regions; and 2) *Sparse supervisory signals*, where certain objects or regions are visible in only a few training views, resulting in weak feature learning and poor generalization across the scene. To address them, we introduce an occlusion-aware semantic scene reconstruction pipeline (Fig. 3).

### 3.1.1 OCCLUSION-AWARE SEMANTIC SCENE RECONSTRUCTION

Let $\mathcal{G} = \{\mathcal{G}_i\}_{i=1}^N$ denote reconstructed 3D Gaussian set, we aim to augment each $\mathcal{G}_i$ with a learnable, $C$-dimensional feature vector $\boldsymbol{f}_i \in \mathbb{R}^C$. The core idea is an association between 2D and 3D (Fig. 3): we first

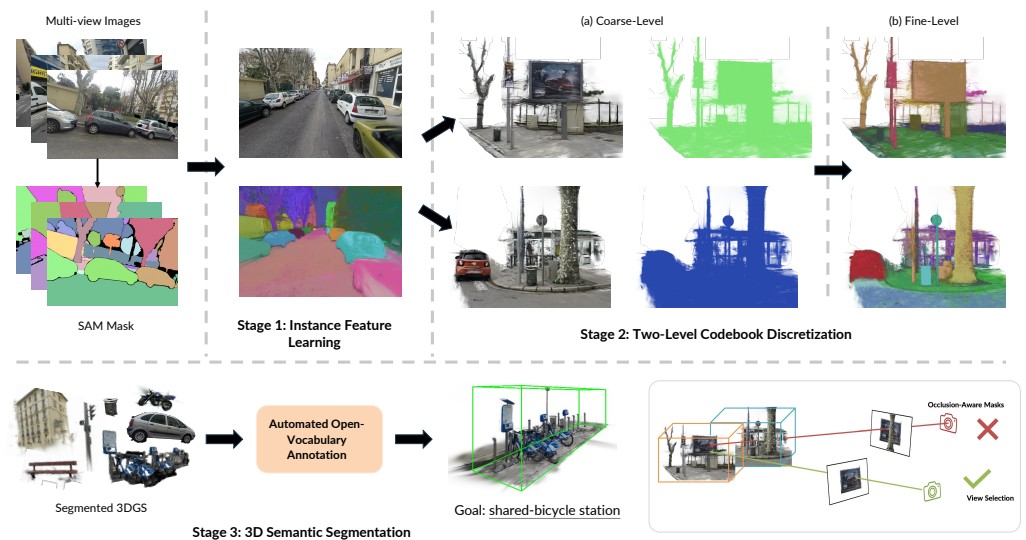

Figure 3: **Occlusion-Aware Semantic Scene Reconstruction.** We achieve occlusion-aware semantic feature learning via 2D-3D association to obtain semantic annotations for building interactive environment.

obtain accurate 2D instance segmentations from multi-view images, then lift these 2D masks into a consistent 3D feature space, and subsequently project the 3D features back into the 2D views to refine segmentation to optimize its feature. This process tackles the issues from four aspects: *instance discrimination*, *instance-level consistency*, *occlusion handling*, and *open-vocabulary annotation*.

**Semantic Scene Feature Learning.** As shown in stage-1 in Fig. 3, we first leverage 2D instance masks from Segment Anything Model (SAM; (Kirillov et al., 2023)) to guide feature learning. Then, for an arbitrary training view, we render per-Gaussian features $\boldsymbol{f} = \{\boldsymbol{f}_i\}_{i=1}^N$ into a 2D feature map with $H$ and $W$ image height and width: $\boldsymbol{F} \in \mathbb{R}^{H \times W \times C}$. Given $K$ binary instance masks $\{\boldsymbol{B}_k\}_{k=1}^K$ predicted by SAM for this view, we enforce feature consistency via contrastive learning (Fig. 3, stage-1): an intra-mask smoothing loss encourages features within a single instance mask to be similar to their mean feature $\bar{\boldsymbol{F}}_k$: $\mathcal{L}_s = \sum_{k=1}^K \sum_{p \in \boldsymbol{B}_k} \|\boldsymbol{F}(p) - \bar{\boldsymbol{F}}_k\|^2$, where $\bar{\boldsymbol{F}}_k = \frac{\sum_{p \in \boldsymbol{B}_k} \boldsymbol{F}(p)}{|\boldsymbol{B}_k|}$. A inter-mask contrastive loss pushes mean features of different instances apart to enhance their distinctiveness: $\mathcal{L}_c = \frac{1}{K(K-1)} \sum_{k=1}^K \sum_{j=1, j \neq k}^K \frac{1}{\|\bar{\boldsymbol{F}}_k - \bar{\boldsymbol{F}}_j\|^2}$, where $\boldsymbol{F}(p)$ is feature vector at pixel $p$. This process produces view-consistent and distinct instance features directly from 2D supervision.

**Coarse-to-fine Codebook Discretization.** To ensure all Gaussians belonging to the same 3D object share an identical instance feature, we use a coarse-to-fine codebook discretization strategy. For coarse level, we cluster the Gaussians using both their learned instance features $\boldsymbol{f}$ and 3D coordinates $\boldsymbol{x}$. Within each coarse cluster, we perform a fine-level quantization using only the instance features $\boldsymbol{f}$. This step identifies distinct object instances within each geometric chunk (Fig. 3, stage-2). To train the model to produce these discrete features, we optimize the quantization loss $\mathcal{L}_q = \|\hat{\boldsymbol{F}} - \boldsymbol{F}_q\|_1$, enforcing feature map $\hat{\boldsymbol{F}} \in \mathbb{R}^{H \times W \times C}$, rendered from the original continuous features learned in Stage 1, matches the feature map $\boldsymbol{F}_q \in \mathbb{R}^{H \times W \times C}$, rendered from the newly quantized features.

**Occlusion-Aware Masks and View Selection.** When training a specific cluster at the fine level, naively rendering its Gaussians in isolation neglects occlusions and context from other objects in the scene, leading to fragmented and inaccurate segmentation. To resolve this, we introduce two mechanisms: 1) *Occlusion-Aware*

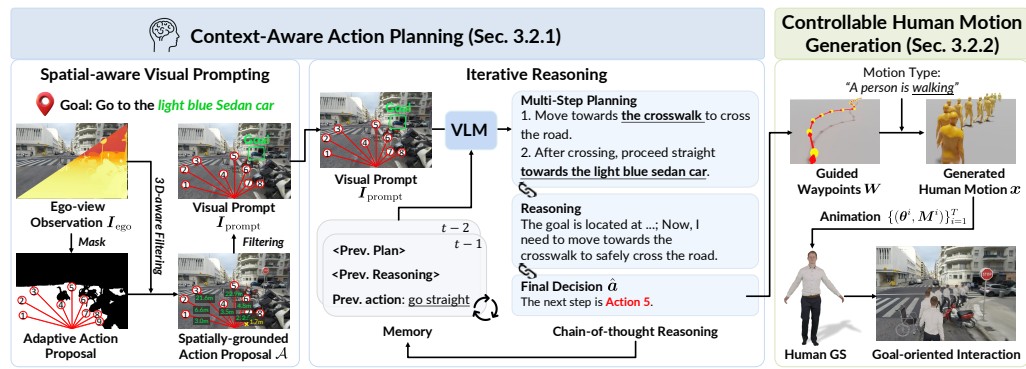

Figure 4: Our **Visually Grounded Humanoid Agent** is designed with a two-level framework of: (1) A **context-aware action planning module** that serves as a high-level planner. It selects actions based on ego-centric visual observations; (2) A **controllable motion generation module** that acts as a low-level controller. It converts the abstract command from the planner into waypoints, which then condition a motion diffusion model to synthesize realistic motion.

*Cluster Masks*: For each target cluster being trained, we perform a rendering pass (Fig. 2, stage-2) in which all other clusters are treated as depth-only occluders. This generates an occlusion-aware, 2D binary mask $\hat{B}_c$ that isolates the truly visible regions of the target cluster from a given viewpoint. The quantization loss $\mathcal{L}_q$ is then applied only within this mask, preventing feature contamination from occluding objects. 2) *Strategic View Selection*: To address the sparse supervision issue, we pre-compute a visibility score for each cluster-view pair (accounting for occlusion) and train a cluster only on views where its projected visible pixel count exceeds a threshold $\delta$ (Fig. 2, stage-2). This focuses the optimization on high-quality views that provide reliable supervision, which both accelerates convergence and improves segmentation accuracy. This dual strategy ensures that feature updates are both accurate, by preventing contamination from occluders, and efficient, by focusing on high-quality views, as demonstrated in our ablation studies (Sec. 4.3).

**Automated Open-Vocabulary Annotation.**    After segmenting 3D instances, we automate textual annotation using Qwen2.5-VL (Bai et al., 2025). We denote a visual prompting mechanism to provide richer descriptions, as detailed in the appendix. Now the resulting scene is photorealistic, semantically rich, and directly supports 3D bounding box extraction for tasks such as goal-based navigation in our benchmark.

### 3.1.2   ANIMATABLE HUMAN AVATAR RECONSTRUCTION AND PLACING

Given human videos captured from the real world, we reconstruct $M$ animatable human avatars, each modeled as a collection of Gaussian primitives. Specifically, the $j$-th human (where $j \in 1, \ldots, M$) at physical timestep $t$ is represented by a set of $N_h$ Gaussians: $\mathcal{G}^{h_j}(t) = \{\mathcal{G}_k^{h_j}(t)\}_{k=1}^{N_h}$, where $\mathcal{G}_k^{h_j}(t)$ denotes the $k$-th Gaussian primitive of the $j$-th human at time $t$. The collection of all dynamic human Gaussians in the scene at time $t$ can thus be written as: $\mathcal{G}^h(t) = \bigcup_{j=1}^{M} \mathcal{G}^{h_j}(t)$. These avatars are randomly placed into the reconstructed scene to populate it with diverse human–scene interactions. Their behaviors are then driven by the **Agent Layer**.

### 3.2   THE AGENT LAYER - VISUALLY GROUNDED HUMANOID AGENTS

To enable *autonomous* human-scene interaction, we design our agent based on a two-level framework that mimics the cognitive paradigm of "slow thinking" for planning and "fast execution" for human motion control, as shown in Fig. 4. In the following, we detail the design of each component.

### 3.2.1   CONTEXT-AWARE ACTION PLANNING

For the high-level planner, we leverage the powerful reasoning capability of VLMs in a training-free, zero-shot way, but constrain the inputs, reasoning chain, and outputs with spatial visual prompting and iterative refinement to ensure grounding in first-view perception and alignment with low-level action execution. Thus,

we formulate the high-level planning problem as a selection task over a discrete set of human-centric action primitives (Nasiriany et al., 2024). This circumvents the inherent limitations of VLMs (Rahmanzadehgervi et al., 2024) in regressing continuous control signals and performing spatial reasoning.

**Action Selection via Visual Prompting.** At each decision step, the agent receives an ego-centric RGB-D observation $I_{\text{ego}} \in \mathbb{R}^{H \times W \times 4}$ and a high-level task description $\mathcal{T}$. We define a discrete action space $\mathcal{A}$ composed of $J$ action primitives: $\mathcal{A} = \{a_1, a_2, \ldots, a_J\}$. Each action primitive $a_j = (\tau_j, d_j)$ is a tuple containing: 1) $\tau_j$: A **motion type** prompt, such as "walk" ,"run" or "wave hands to someone" *etc*; 2) $d_j$: A **canonical direction vector** in the 2D image plane, representing a potential direction of next movement.

The VLM is then prompted with the task $\mathcal{T}$ and the visual context $I_{\text{prompt}}$ to select the optimal action primitive $\hat{a}$. A straightforward approach is to overlay a fixed set of directional arrows onto the agent's ego-centric view. However, such a strategy is agnostic to the underlying 3D scene geometry and traversability, which can mislead the VLM into selecting physically impossible or unsafe paths (see our ablation in Sec. 4). The challenge is magnified in large-scale outdoor environments, where goals are often distant, requiring long-horizon planning, and the agent's view is frequently cluttered with complex objects that can occlude the target during navigation (as shown in Fig. 3).

To this end, we introduce a pipeline with two key innovations: 1) **Spatial-Aware Visual Prompting**, which *discretizes* and *emphasizes* the navigation space into several candidate directions and grounds them with 3D spatial, semantic, and goal context, enabling the VLM has a purposeful spatial-aware visual input $I_{\text{prompt}}$ to select a physically plausible one-step action; and 2) **Iterative Reasoning**, which *continues* planning across multiple steps by maintaining memory and reasoning chains, allowing robust long-horizon navigation, obstacle avoidance, and handling of target occlusion.

While spatially-aware visual prompting enhances *single-step* decision-making, it can encourage myopic strategies that are suboptimal for *long-horizon* goals (*e.g.*, turning directly toward a goal instead of first navigating around an obstacle, as shown in Fig. 5). To endow the agent with foresight, we introduce an iterative reasoning mechanism inspired by Chain-of-Thought (CoT) prompting (Wei et al., 2022), but further adapt it to take visual information into consideration in our setting. Concretely, we reframe the task as a continuous planning process. At each timestep $t$, the text prompt is dynamically updated to include: 1) the high-level language goal, 2) a memory buffer with the plan, reasoning, and actions from previous steps $t-1, t-2, etc.$, and 3) a query asking the VLM to update its plan and select the next action based on the current observation. The VLM's output is a tuple containing an updated natural language plan and the index of the chosen action $\hat{k}$. This iterative loop enables the agent to balance long-term strategy with immediate visual feedback, reducing collisions by integrating foresight into embodied decision-making. Crucially, this mechanism also handles target occlusion. By leveraging the initial goal localization, the agent can detect if the target is no longer in view. If so, it enters a recovery mode, relying on its memory and existing plan to navigate around the obstacle until the target is reacquired. This effectively reduces failures in complex environments where the goal is not always visible.

### 3.2.2 CONTROLLABLE HUMAN MOTION GENERATION

Given a high-level action primitive $\hat{a} = (\hat{\tau}, \hat{d})$ produced by the planner, the low-level controller generates a continuous, full-body trajectory. We leverage the 3D target point $\hat{p}$ computed by the planning module and construct a sequence of $N_w$ waypoints $W = \{w_1, \ldots, w_{N_w}\}$ by linearly interpolating between the agent's current position and $\hat{p}$. These waypoints, along with the motion type text $\hat{\tau}$ (*e.g.*, "walk"), condition a diffusion model to synthesize a plausible full-body motion sequence $x = \{x^i\}_{i=1}^T$, where each $x^i \in \mathbb{R}^d$ represents a single pose. Note that we factor out the global transformation $M = \{M^i\}_{i=1}^T$, first to condition the motion generation model to operate in a canonical coordinate frame. This process is illustrated in Fig. 4.

**Training-Free Guidance.** We employ a kinematic-based generative model for motion synthesis, prioritizing pose naturalness and controllability over RL-based methods (Rempe et al., 2023; Wang et al., 2024; Wei et al.,

2024). Specifically, we adapt MDM-SMPL (Petrovich et al., 2024), a diffusion model that generates SMPL parameters, by incorporating a training-free guidance mechanism (Liu et al., 2024a; Karunratanakul et al., 2024). This allows us to steer the generation process at inference time. Each denoising update uses a weighted sum of loss terms that enforce consistency with the text prompt $\hat{\tau}$, adherence to the waypoints $\boldsymbol{W}$, and temporal smoothness. The guidance step at the diffusion timestep $k$ is formulated as: $\tilde{\boldsymbol{x}}_k = \boldsymbol{x}_k - \alpha \nabla_{\boldsymbol{x}_k} \mathcal{L}(\boldsymbol{x}_k; \boldsymbol{W}, \hat{\tau})$, where $\alpha$ is the guidance scale and $\mathcal{L}$ is the combined loss. The final motion sequence $\boldsymbol{x}$ is converted to SMPL parameters $\{(\boldsymbol{\theta}^i, \boldsymbol{M}^i)\}_{i=1}^T$ for animation. See the appendix for more details.

**Goal-oriented Interaction.** After obtaining the motion sequence $\boldsymbol{x}$ as SMPL parameters, we retarget it to drive the animatable GS-based human avatar embedded in the reconstructed scene. The avatar moves through the scene following the generated trajectory, interacting with the environment via mesh-based collision handling and ground-contact constraints provided by the World Layer. This integration allows the agent to physically embody its planned actions, enabling goal-directed behaviors such as approaching a landmark, avoiding obstacles, or reaching a specified target. Importantly, the outcome of each executed sequence is fed back into the perception stream, closing the loop between motion generation and high-level planning.

# 4 EXPERIMENTS

## 4.1 BENCHMARK DESIGN

**Datasets.** We utilize large-scale *SmallCity* street scene from hier-GS (Kerbl et al., 2024) dataset, covering a $100m \times 100m$ outdoor area spanning four city blocks. To populate the environment, we apply GaussianAvatar (Hu et al., 2024) to reconstruct animatable GS-based avatars of 6 identities: four from PeopleSnapShot dataset (Alldieck et al., 2018a), and two from GaussianAvatar dataset.

**Task design.** As there are no direct baselines for comparing the full world–agent paradigm, we focus our benchmark on navigation task, where comparison is most meaningful to demonstrate autonomy. We design two goal-oriented visual navigation tasks, *SimNav* and *ObstNav*. **Level-1 (SimNav)** evaluates agent's fundamental goal-reaching ability. The agent is placed in an environment with a generally clear path to the destination, free of major obstacles. **Level-2 (ObstNav)** assesses an agent's ability to reason and adapt its plan, which is more complex. A static obstacle is placed along the direct path between agent's starting point and goal, requiring the agent to balance goal-reaching with collision avoidance. To focus evaluation on high-level planning rather than motion pattern, agent's motion type is fixed to "walking".

**Baselines.** We compare our method against two SOTA visual-language navigation approaches: NaVid (Zhang et al., 2024a) and NaVILA (Cheng et al., 2024). They employ a two-level architecture where a VLM generates intermediate language commands for a low-level locomotion controller. To adapt these baselines to our framework, we intercept their mid-level language commands and convert them into waypoints that guide our motion generation model, ensuring a fair comparison of high-level planning capabilities. The query frequency is kept consistent across all methods.

**Metrics.** The performance of the humanoid agent on each task is evaluated based on the success rate (SR (Anderson et al., 2018)), the success rate weighted by path length (SPL (Batra et al., 2020)), and the collision rate (CR). To account for the stochastic nature of VLM-based decision-making, all experiments are conducted three times with different random seeds, and we report the average results.

## 4.2 EXPERIMENTAL RESULTS

We report quantitative results for both tasks in Tab. 2. Our method consistently outperforms two state-of-the-art VLN approaches by a substantial margin. In the SimNav task, it achieves the highest SR, SPL, and CR, with an SR of 68.3%, exceeding the strongest baseline (NaVid) by 30.9%. In the more challenging ObstNav task, our method also yields significantly higher SR and SPL, while its CR is slightly lower than that of NaVid. This minor drop stems from our strategy's emphasis on balancing goal reaching and collision avoidance, which can occasionally increase collisions in crowded environments in order to successfully reach the goal. Finally, qualitative results in Fig. 5 demonstrate our VLM-based agent's capacity for complex reasoning, showing it can bypass obstacles and navigate to distant destinations that require long-horizon planning.

Table 1: **Quantitative comparison.** Our method significantly outperforms two SOTA VLN approaches on both tasks, achieving a better balance between reaching the goal and avoiding collisions.

| Task | SimNav | | | ObstNav | | |
|---|---|---|---|---|---|---|
| Method | SR ↑ | SPL ↑ | CR ↓ | SR ↑ | SPL ↑ | CR ↓ |
| NaVILA (Cheng et al., 2024) | 22.5% | 0.199 | 70.8% | 20.8% | 0.176 | 75.0% |
| NaVid (Zhang et al., 2024a) | 37.4% | 0.279 | 19.2% | 32.5% | 0.233 | **23.1%** |
| Ours | **68.3%** | **0.640** | **13.3%** | **55.8%** | **0.516** | 30.8% |

Table 2: **Ablation study of the VLM-based planning paradigm.** The best results are in **bold** and the second-best are underlined.

| Method | | SimNav | | | ObstNav | | |
|---|---|---|---|---|---|---|---|
| Visual Prompting | Iterative Reasoning | SR ↑ | SPL ↑ | CR ↓ | SR ↑ | SPL ↑ | CR ↓ |
| ✗ | ✗ | 53.3% | 0.519 | 46.7% | 38.3% | 0.365 | 60.0% |
| ✓ | ✗ | 57.5% | 0.501 | 32.5% | 46.7% | 0.395 | 47.5% |
| ✗ | ✓ | 49.2% | 0.445 | **7.6%** | 44.2% | 0.387 | **17.7%** |
| ✓ | ✓ | **68.3%** | **0.640** | 13.3% | **55.8%** | **0.516** | 30.8% |

## 4.3 ABLATION STUDIES

**Spatial-Aware Visual Prompting.** We evaluate the effectiveness of spatial-aware visual prompting via an ablation study. As shown in Tab. 2, incorporating this component improves the SR across both tasks, with particularly notable gains on the more challenging ObstNav task, due to the prompts' ability to emphasize object locations. This mechanism also complements iterative reasoning. For instance, in Fig. 5, a VLM might generate a coarse initial plan for a distant goal. Without visual prompting, the agent follows this outdated plan even as it nears the target's vicinity, ultimately failing to locate it (see the first and the third row). In contrast, our spatially-aware prompts inject strong semantic cues about the goal's position, urging the agent to update its plan and correct for initial inaccuracies. By enabling the agent to dynamically adapt and better reconcile conflicts between textual and visual information, this mechanism boosts SR by over $10\%$ on both tasks.

**Iterative Reasoning.** We further investigate the impact of the iterative reasoning mechanism. As shown in Tab. 2, integrating iterative reasoning on top of our visual prompting strategy yields substantial improvements across three evaluation metrics. The observed gains in SR and SPL indicate that iterative reasoning effectively facilitates long-horizon goal achievement and prevents myopic decision-making (row two in Fig. 5). Additionally, iterative reasoning significantly reduces CR, as agents tend to navigate around obstacles more carefully during the planning. Interestingly, applying iterative reasoning without visual prompting decreases SR and SPL while significantly improving CR. This occurs because the agent becomes overly cautious to avoid collisions but, lacking visual prompts to keep the target salient, loses track of its goal during the multi-step reasoning process (Fig. 5). This highlights the strong complementarity between our two components.

## 5 CONCLUSION

In this work, we introduced Visually-grounded Humanoid Agents, a coupled two-layer paradigm for embodied digital humans, consisting of a world layer that reconstructs semantically enriched 3D environments with animatable Gaussian avatars and an agent layer that governs their behavior through perception–action loops. Looking ahead, we plan to extend this framework to richer embodied skills, such as conversations and interaction with scenes, and applications in robotics, such as human-centric robotic learning.

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

# A IMPLEMENTATION DETAILS

## A.1 SEMANTIC SCENE RECONSTRUCTION

**Large-scale Scene reconstruction.** Following Hier-GS (Kerbl et al., 2024), we partition the *SmallCity* scene into 4 spatially adjacent blocks ($2 \times 2$ layout), each reconstructed in parallel with standard pipeline, and subsequently fused to form the complete scene. To enhance geometric accuracy, we incorporate two regularization techniques. Inspired by Huang et al. (2024), we regularize the 3D Gaussians towards 2D disks to improve the geometric details. Additionally, following (Kerbl et al., 2024), we leverage monocular depth priors estimated from (Yang et al., 2024c) to guide the optimization, which significantly improves visual quality, particularly for the reconstruction of flat road surfaces.

**Mesh extraction.** For the mesh extraction, we first render the depth map from each frame and fuse them into a TSDF field with Open3D (Zhou et al., 2018), the mesh is then extracted from TSDF field with voxel size set to 0.2; depth truncation = 60m. To eliminate the potential dynamics issue caused by uneven ground reconstructions, we further remove the ground plane of the scene in our mesh extraction through ground plane segmentation. We observed that general-purpose segmentation models like SAM (Kirillov et al., 2023) struggled to consistently isolate the entire ground surface due to prompt ambiguity. Consequently, we perform a manual filtering step to ensure all ground-plane mesh is correctly removed.

**SAM mask generation.** Processing all fine-grained SAM masks is computationally prohibitive for large-scale scenes. Therefore, we implement a filtering strategy to select high-quality masks. We discard masks that are either too small, exhibit high internal variance (indicating a potentially noisy segmentation), or correspond to dynamic objects. This pre-processing step curates a stable and meaningful set of instance masks for the annotation stage.

**Automated Open-Vocabulary Annotation.** Once 3D instances are segmented, the final step is to assign them meaningful textual annotations. Unlike prior works such as OpenGaussian (Wu et al., 2024b) which require predefined text prompts to query objects (a "text-to-instance" task), our objective is the inverse: to automatically generate rich descriptions for given instances (an "instance-to-text" task).

To fully automate the semantic annotation process, we leverage Qwen2.5-VL Bai et al. (2025), as an automated annotator. For each segmented 3D instance, we first select a set of $K = 5$ optimal camera views. The selection is guided by a scoring function that considers both the Intersection over Union (IoU) with the 2D SAM masks and the instance's visibility ratio within the frame. This score explicitly accounts for occlusion, ensuring the VLM is queried with views where the object is clearly visible. We employ a visual prompting technique where we highlight the object's contour in the image while slightly dimming the background. This focuses the VLM's attention while preserving contextual cues (e.g., "the motorcycle near the sign"), leading to richer and more accurate descriptions of object appearance, material, and purpose. The captions generated from the $K$ views are then consolidated by an LLM (Achiam et al., 2023) into a single, cohesive annotation.

## A.2 CONTEXT-AWARE ACTION PLANNING

We query the VLM for a planning update every $T = 20$ simulation steps, corresponding to approximately 1 Hz, which mimics a natural human decision-making frequency.

## A.3 CONTROLLABLE HUMAN MOTION GENERATION

As discussed earlier, the high-level planning module yields action commands and sparse guiding signals (*i.e*, motion prompt, trajectory, orientation, previous poses for alignment). Although training-based controllable

motion generative approaches have been developed to enable spatial control signals over any joint Xie et al. (2023); Wan et al. (2024); Shafir et al. (2023), the output format of motion is not directly compatible with SMPL Loper et al. (2023); Pavlakos et al. (2019) parameters, requiring expensive test-time optimization to be converted into SMPL poses that can be used to animate the avatars Hu et al. (2024). This bottleneck becomes inneglectable as we aim at synthesize long motion sequences for multiple agents simultaneously.

**Sampling Strategy.** MDM-SMPL Petrovich et al. (2024), explain the pose representation here; different from HumanML3D; also, we sample $T = 100$ steps and only apply training-free spatial guidance starting from $t_0 = 0.5$ (normalized timestep), and the guidance scale factor is 5.0.

### A.4 NAVIGATION BENCHMARK AND TASK DESIGN

We design a challenging navigation benchmark to evaluate performance. A task is considered successful if the agent reaches within a distance of d=1.0m of the target. To evaluate path efficiency, we use the Success weighted by Path Length (SPL) metric. The optimal path length required for SPL is pre-computed for all scenarios using the A* algorithm on a 2D Bird's-Eye-View (BEV) occupancy map with a grid resolution of 0.2m.

For our test set, we designate 40 distinct landmarks (e.g., "crosswalk," "bicycle station") within the reconstructed scene. For each landmark, we define its corresponding 3D bounding box as the target region. An agent is initialized at a random position between 6 and 15 meters from the landmark, with its initial orientation facing the goal. We generate 5 random initial states for each of the 40 landmarks, creating a total of 200 distinct test scenarios.

### A.5 BASELINES

We compare our method against state-of-the-art, training-based embodied navigation agents, including NaVILA (Cheng et al., 2024) and NaVid (Zhang et al., 2024a). Since these methods generate high-level language commands (*e.g.*, "walk forward towards the crosswalk"), we adapt them to our framework by converting their output into a guided trajectory for the motion generation module. To ensure a fair comparison, we standardize the frequency at which the Vision-Language Model (VLM) is queried across all methods. For all approaches, including our own, the policy takes a history of the previous 8 frames as input.

### A.6 ADDITIONAL EXPERIMENTAL RESULTS

**ABlation Study of Occlusion-Aware masks and view selection.** We first evaluate the effectiveness of our occlusion-aware semantic reconstruction pipeline. As shown in Fig. 6, the key insights are two aspects: 1) Occlusion-Aware mask significantly improves the consistency and density of instance features. With masks, each instance exhibits compact, contiguous features, whereas without them, the projected features contain many holes and inconsistencies due to occluders contaminating the signal. This highlights the importance of explicitly accounting for occlusion during training. 2) By prioritizing views where the target instance is sufficiently visible, our method successfully recovers thin or sparsely observed objects (*e.g.*, bins, traffic signs) that are often missed and not segmented under naive training. Together, these two techniques yield semantically cleaner and more complete reconstructions, forming a stronger foundation for downstream humanoid–scene interaction.

### A.7 ADDITIONAL RELATED WORK

**3D Gaussian Splatting** 3D Gaussian Splatting (3DGS; (Kerbl et al., 2023)) is a state-of-the-art technique that represents scenes with 3D Gaussian primitives to achieve photorealistic rendering. Each Gaussian $\mathcal{G}_i$

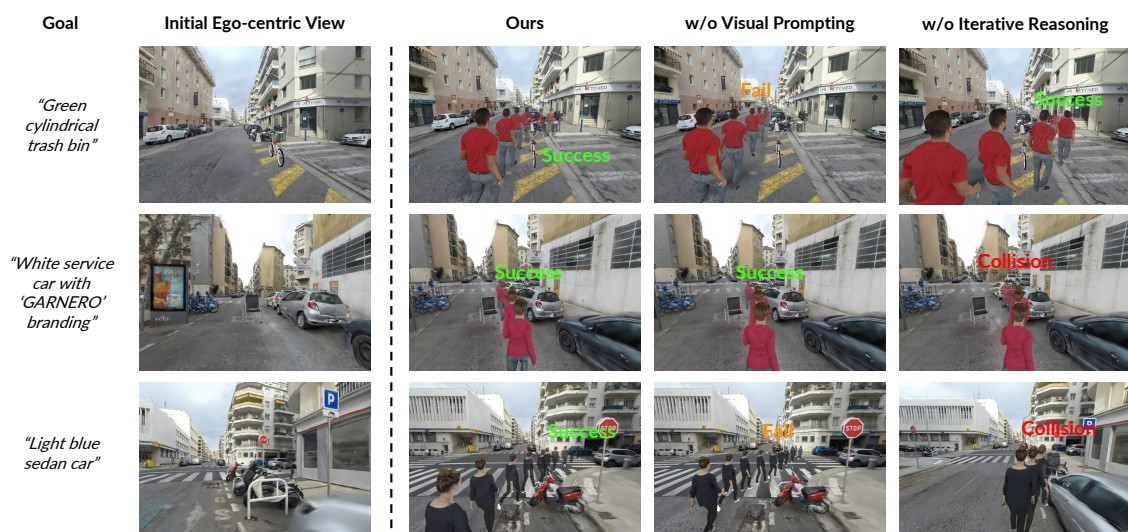

Figure 5: **Qualitative results of the ablation study on the VLM-based planning paradigm.** We compare our full model with variants that remove spatial-aware visual prompting or iterative reasoning. The results demonstrate that the variant lacking spatial grounding is prone to making navigational errors, causing it to deviate from the optimal path and lose track of the goal, often resulting in failure. The variant without iterative reasoning exhibits short-sighted behavior; it tends to follow simplistic, straight-line paths and fails to perform complex planning, leading to a significantly higher collision rate. In contrast, our full model produces robust, goal-directed trajectories that successfully navigate around obstacles.

is parameterized by a set of learnable attributes: a mean (center) $\boldsymbol{\mu}_i \in \mathbb{R}^3$, a covariance matrix $\boldsymbol{\Sigma}_i \in \mathbb{R}^{3\times3}$, an opacity $o_i \in (0,1)$, and a view-dependent color stored as Spherical Harmonics (SH) coefficients $\boldsymbol{c}_i$. The influence of a Gaussian at any 3D point $\boldsymbol{x}$ is given by:

$$\mathcal{G}_i(\boldsymbol{x}) = \exp\left(-\frac{1}{2}(\boldsymbol{x} - \boldsymbol{\mu}_i)^T \boldsymbol{\Sigma}_i^{-1}(\boldsymbol{x} - \boldsymbol{\mu}_i)\right). \tag{1}$$

To render an image, these 3D Gaussians are projected onto the 2D camera plane. For each pixel $\boldsymbol{p}$, the final color $\mathbf{C}$ is then determined by sorting all overlapping Gaussians by depth and accumulating their contributions through standard alpha-blending:

$$\mathbf{C}(\boldsymbol{p}) = \sum_{i \in N} T_i \boldsymbol{c}_i \alpha_i, \quad \text{where} \quad T_i = \prod_{k=1}^{i-1}(1 - \alpha_k). \tag{2}$$

Here, $\alpha_i = o_i \mathcal{G}'_i(\mathbf{p})$ is the alpha contribution of the $i$-th Gaussian, determined by its opacity and the projected 2D Gaussian function $\mathcal{G}'_i$. This fully differentiable rasterization can similarly be used to compute other attributes, such as the per-pixel depth map. Furthermore, a key advantage of this explicit representation is that rigid transformations can be directly applied to the Gaussians by updating their mean and rotation parameters, making them well-suited for modeling dynamic objects.

**Animatable Gaussian-based Human Avatar** Recent works have extended 3DGS to model animatable human avatars (Hu et al., 2024; Lei et al., 2024; Chen et al., 2024). These approaches represent the avatar in a canonical space (*i.e.*, a standard T-pose) with a set of $N_h$ Gaussians, denoted as $\hat{\mathcal{G}}^h = \{\hat{\mathcal{G}}^h_k\}_{k=1}^{N_h}$. This

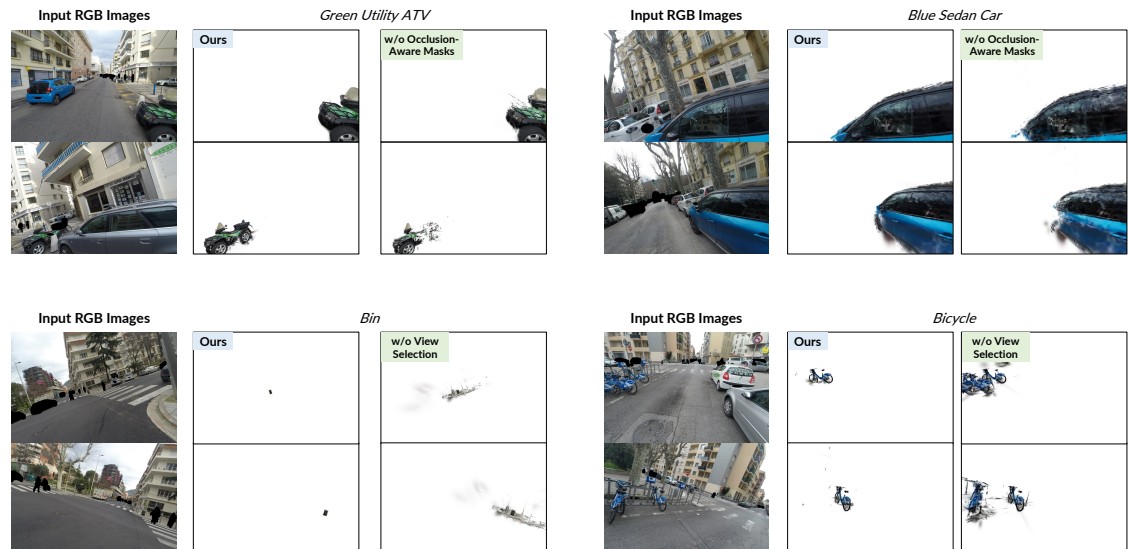

Figure 6: **Ablation study on the semantic scene reconstruction paradigm.** Our framework achieves precise 3D instance segmentation with well-defined boundaries, demonstrating robustness to severe occlusion while successfully recognizing thin or small objects in large-scale outdoor scenes.

canonical model is then deformed into a posed space using animation controls from the parametric body model SMPL (Loper et al., 2023; Pavlakos et al., 2019). At a given timestep $t$, this deformation is driven by the static body shape $\boldsymbol{\beta}$, time-varying pose parameters $\boldsymbol{\theta}(t)$, and a global rigid transformation $\boldsymbol{M}(t) \in SE(3)$. The final posed Gaussians $\mathcal{G}^h(t)$ are obtained by applying the standard Linear Blend Skinning (LBS) function to the canonical Gaussians:

$$\mathcal{G}^h(t) = \boldsymbol{M}(t) \circ \text{LBS}(\hat{\mathcal{G}}^h, \boldsymbol{\theta}(t), \boldsymbol{\beta}) \tag{3}$$

where $\circ$ denotes the application of the global transformation to the posed avatar.

Building on these advances in scene and human modeling, we propose a unified framework that integrates a static 3DGS scene $\mathcal{G}^s$ with $M$ animatable Gaussian-based avatars, $\mathcal{G}^h(t) = \{\mathcal{G}^{h_j}(t)\}_{j=1}^M$. Crucially, to support intelligent interaction, we enrich this unified representation with semantic information. The resulting semantically-aware world serves as the foundation for agents to perceive and act, as detailed in the following section.

## A.8 USE OF LLM

We used LLM in two ways: **1)** to polish the manuscript by correcting grammar and improving writing fluency, and **2)** to generate textual annotations for 3D bounding boxes during semantic scene reconstruction (as mentioned in Sec 3.1.1 "Automated Open-Vocabulary Annotation" part).

## B LIMITATIONS

As the first baseline for embodied digital humans in reconstructed 3D environments, our framework inevitably has limitations that point to promising future directions. **(1)** Reconstruction artifacts remain an issue, as

limited camera field-of-view can cause smaller obstacles to disappear when approached. **(2)** The system currently lacks a unified, end-to-end trained policy, instead relying on modular components for perception, planning, and action. **(3)** Our evaluation is restricted to navigation with simple walking behaviors, leaving more complex and interactive human motions (e.g., collaboration or social interaction) to future work. **(4)** Finally, while we focus on digital humans in simulation, extending this paradigm to real-world deployment offers a pathway to robot learning, where reconstructed scenes and agent behaviors can inform humanoid robots in embodied AI settings.

