# OpenReview forum: "Visually-grounded Humanoid Agents"
_ICLR.cc/2026/Conference — ICLR 2026 Conference Withdrawn Submission_

### Official Review · Reviewer_XrfH · 2025-10-28

**Soundness:** 3
**Presentation:** 3
**Contribution:** 3
**Rating:** 6
**Confidence:** 3

**Summary:**

The paper introduces Visually-grounded Humanoid Agents, a framework to create autonomous digital humans that can actively perceive, reason, and act in novel 3D scenes reconstructed from real-world videos. The paper addresses the limitations of existing passive, scripted digital humans by proposing a coupled world / agent paradigm. The World Layer reconstructs large-scale 3D environments using an occlusion-aware semantic 3D Gaussian Splatting pipeline and populates them with animatable avatars . The agent layer transforms these avatars into autonomous agents by equipping them with first-person perception and a perception-action loop. This layer acts as an high-level planner, which uses a VLM in a zero-shot manner, constrained by a spatial-aware visual prompting ground decisions in 3D geometry and iterative reasoning to provide memory for long-horizon planning and obstacle avoidance . A low-level diffusion model then translates these plans into full-body motion. The authors validate their system on a new navigation benchmark, demonstrating significantly higher task success rates and fewer collisions compared to SOTA baselines like NaVid and NaVILA . Ablation studies confirm that the visual prompting and iterative reasoning components are both critical to the agent's robust performance .

**Strengths:**

1. The paper introduces a comprehensive two-layer (world and agent layers) paradigm that does 3D scene reconstruction and provides autonomous digital avatar movement toward a goal. It also seems that this is the first paper to propose digital human navigation in reconstructed 3D environments as previous methods focus on robotics.
2. The paper presents a novel “occlusion-aware semantic scene reconstruction pipeline” to build 3DGS scenes. The method is specifically designed to handle major challenges in large-scale urban environments, such as severe occlusion and sparse supervisory signals.
3. The "Agent Layer" presents a solution to the common failings of VLMs in embodied tasks. Instead of naively using a VLM, it introduces Spatial-aware Visual Prompting that grounds the VLM's decisions in the 3D geometry and semantics of the scene, preventing it from selecting physically impossible paths.
4.The paper proposes a new benchmark for evaluating humanoid-scene interaction within reconstructed 3D environments, complete with two distinct tasks (SimNav and ObstNav) and evaluates it on clear metrics (SR, SPL, CR).
5. The proposed method yields better results towards two state-of-the-art VLN baselines on the paper's navigation tasks.

**Weaknesses:**

1. Privileged Sensory Input: The agent uses ego-centric RGB-D observation. In a simulated world reconstructed from 3DGS, the depth channel is likely rendered perfectly from the 3D model. This is a form of privileged, noise-free information that is not available in the real world, where depth sensors like LiDAR or ToF cameras produce sparse and noisy data. This creates a significant perception gap for any real-world deployment. There is a big lack of clarification here from where the depth information comes from, is it derived directly from the 3DGS or if a depth model (e.g Depth Anything, Marigold, etc…) is being used.
2. Significant Computational Cost and Latency: The proposed agent layer appears to be extremely computationally expensive, which undermines the claim of enabling agent population "at scale". Since the scene  is populated with many animatable human avatars, I suppose that each avatar has its own instance of VLM. Do they all run at the same time or are they the results of different generations ? Running a VLM can be quite expensive and it is not clear which one is being used, QwenVL-2.5 is mentioned for captioning. Is this the model that is also used for the agent layer ? What’s the size of the VLM ? The same concern hold for the motion generation.
3. Limited Evaluation and Generalization Claims: The paper's core claim is to enable agents in "novel, unseen environments". However, the entire experimental benchmark is built on a single large-scale scene (the "SmallCity" dataset). The "SimNav" and "ObstNav" tasks only test generalization to new goals and paths within this one environment. There is no evidence that the system (particularly the complex world reconstruction pipeline) can generalize to truly novel environments with different architectures, object densities, or lighting like in indoor context (like navigation in a house or a mall for example). I’m afraid that the vast space offered by the city setting is “too easy” for the model to navigate in it and I would like to see how it behaves in a more challenging setting for the benchmark.
4. Missing "Human-Likeness" Evaluation: The paper repeatedly claims to create agents that "behave like real people" and exhibit "natural" behaviors. However, the evaluation metrics are standard robotics metrics: Success Rate (SR), Success weighted by Path Length (SPL), and Collision Rate (CR). These metrics measure task completion and efficiency, not the quality or naturalness of the behavior. A human-likeness study (e.g. a user study) is absent, making the "humanoid" and "natural" claims unsupported by the evidence.
5. Prompting details for the VLM: It would’ve been good to have the details about the system prompt for the iterative reasoning in Appendix because this is an important piece for the model behavior.

**Questions:**

see weaknesses.

---

### Official Review · Reviewer_yK7Q · 2025-11-01

**Soundness:** 2
**Presentation:** 3
**Contribution:** 2
**Rating:** 4
**Confidence:** 3

**Summary:**

This paper introduces Visually‑grounded Humanoid Agents, a two‑layer world–agent paradigm for digital humans that look, perceive, reason, and act in reconstructed real‑world 3D scenes using only visual observations and goal descriptions. The World Layer reconstructs large outdoor scenes via 3D Gaussian Splatting with an occlusion‑aware semantic scene reconstruction pipeline that (i) learns instance features with 2D–3D association, (ii) uses coarse‑to‑fine codebook discretization to ensure instance consistency, and (iii) introduces cluster masks and strategic view selection to handle occlusion/sparse views; it then auto‑annotates instances with Qwen2.5‑VL. The Agent Layer performs context‑aware planning with spatially‑aware visual prompting and iterative reasoning, then executes actions via diffusion‑based motion generation. A benchmark on a 100m×100m urban scene evaluates SimNav and ObstNav tasks: the method improves SR and SPL substantially over NaVid and NaVILA, with ablations showing complementary benefits of visual prompting and iterative reasoning.

**Strengths:**

Clear, coherent architecture that closes a perception–action loop in realistic scenes; the two-layer design is well motivated and visually explained (Fig. 2 & Fig. 4).
Occlusion-aware semantic scene reconstruction is a thoughtful contribution to building structured, queryable outdoor worlds from videos.
Planning innovations (spatially-aware visual prompting + iterative reasoning) are simple, training-free, and empirically complementary; ablations are convincing.

**Weaknesses:**

The evaluation scope is narrow—experiments are restricted to one outdoor SmallCity scene with walking-based navigation tasks. The generality of the approach to indoor settings, social multi-agent interactions, or manipulation remains unexplored.

The World Layer leans heavily on engineered 3D reconstruction and rendering rather than novel learning or reasoning. The occlusion-aware semantic reconstruction largely combines existing tools (3DGS, SAM, CLIP/Qwen2.5-VL) in a modular pipeline, resulting in a well-structured but static environment that may overstate the agent’s autonomy.

The agent’s intelligence is modest, limited to discrete navigation decisions under clean RGB-D observations, without demonstrating robustness to perceptual noise, partial occlusions, or dynamically changing scenes.

The automatic 3D instance annotation via Qwen2.5-VL lacks quantified accuracy, and potential open-vocabulary hallucinations or label biases are not discussed.

Only six humanoid avatars are tested; expanding to diverse body shapes, motion types, and multi-agent interactions would better assess robustness and generalization.

**Questions:**

See Weakness

---

### Official Review · Reviewer_nbKP · 2025-11-01

**Soundness:** 2
**Presentation:** 2
**Contribution:** 1
**Rating:** 2
**Confidence:** 3

**Summary:**

The paper proposes a pipeline for creating goal-driven humanoid agents that act inside 3D environments reconstructed from video. It introduces a two-layer architecture: (1) a World Layer that reconstructs static scenes using large-scale 3D Gaussian splatting, performs occlusion-aware instance segmentation, and uses open-vocabulary vision-language labeling to generate semantic label; and (2) an Agent Layer that enables visual-language-driven human behavior. A VLM planner selects high-level action primitives (motion type + direction) using ego-centric RGB-D observations; then a diffusion-based motion generator synthesizes full-body SMPL motions guided by waypoints and text prompts.

**Strengths:**

The paper builds a pipeline that connects large-scale scene reconstruction, semantic labeling, VLM-based planning, and full-body humanoid motion generation.

The idea of grounding digital humanoid agents in reconstructed real-world 3D scenes is interesting and potentially impactful for autonomous driving.

**Weaknesses:**

Novelty is limited. Most modules (Gaussian reconstruction, SAM segmentation, VLM reasoning, motion diffusion model) are based on existing techniques, and the overall contribution is limited.

Evaluation. Results are mostly qualitative or based on navigation metrics. There is little quantitative evaluation of motion quality and realism. The paper also lacks evaluation.

Static-world assumption. The system filters out dynamic or partially occluded elements during reconstruction, leading to simplified static environments. This limits both realism and generality.

Limited behavioral diversity. The evaluation focuses only on navigation-type behaviors; no experiments are provided on other types of interactions, such as object manipulation, multi-agent scenarios, or dynamic goals.

**Questions:**

How well would the VLM planner generalize to unseen or ambiguous language goals, given that the action primitives and prompting strategy are predefined?

---

### Official Review · Reviewer_prwz · 2025-11-01

**Soundness:** 2
**Presentation:** 3
**Contribution:** 3
**Rating:** 6
**Confidence:** 3

**Summary:**

The authors propose a system to simulate large-scale realistic humanoid agents acting realistically in large-scale 3D environments. This consists of two parts: the World Layer, which adopts CityGaussian [1] to simulate large-scale scenes and annotates objects within it, and the Agent Layer, which simulates humanoid avatars [2] acting within these scenes. The Agent Layer is achieved by combining a low-level controller for the motion synthesis (using [3], a diffusion model for SMPL parameters), and a high-level planner which uses a Vision-Language Model (VLM) for its core but adds two techniques, "Spatial-Aware Visual Prompting" (uses 3D information to adjust the VLM prompt) and "Iterative Reasoning" (maintains VLM context over multiple steps).

The high-level planner is tested on language-conditioned visual navigation tasks within this simulation and compared against two baselines from the literature, NaVid [4] and NaVILA [5], validating the effectiveness of the techniques for improving planner performance.

[1] Liu, Yang, et al. "Citygaussian: Real-time high-quality large-scale scene rendering with gaussians." European Conference on Computer Vision. Cham: Springer Nature Switzerland, 2024.
[2] Hu, Liangxiao, et al. "Gaussianavatar: Towards realistic human avatar modeling from a single video via animatable 3d gaussians." Proceedings of the IEEE/CVF conference on computer vision and pattern recognition. 2024.
[3] Petrovich, Mathis, et al. "Multi-track timeline control for text-driven 3d human motion generation." Proceedings of the IEEE/CVF Conference on Computer Vision and Pattern Recognition. 2024.
[4] Zhang, Jiazhao, et al. "Navid: Video-based vlm plans the next step for vision-and-language navigation." arXiv preprint arXiv:2402.15852 (2024).
[5] Cheng, An-Chieh, et al. "Navila: Legged robot vision-language-action model for navigation." arXiv preprint arXiv:2412.04453 (2024).

**Strengths:**

The work on the "World Layer" is particularly strong. A specific strength of this work is the systems integration work to build a compositional 3D environment by combining 3DGS-based large-scale environments with 3DGS-based humanoid avatars, which is nontrivial. The careful work on 3D segmenting objects and and textually annotating them is an important contribution to the literature.

The approach of the "Agent Layer" is quite sensible; by leveraging a VLM and incorporating the available 3D information into the prompt and utilizing memory + Chain-of-Thought for the VLM to take decisions are all very reasonable choices, and in particular naturally extends to future work on improving the realism of the humanoid agents for embodied skills beyond just navigation.

The navigation experiments are executed well and compared against reasonable baselines with standard navigation metrics.

**Weaknesses:**

It feels experiments which justify the set of choices in building the simulation (specifically, section 3.1 in the World Layer) would strengthen the paper significantly. There's significant novelty and carefulness in the choices made in this section and ablations / demonstrations that these improve performance of e.g. the final segmentations / annotations would be very helpful, as in my mind it forms the core contribution of the work. For example, the navigation experiments are done from an egocentric view, so the fact that 3DGS avatars were used and the human motion generation is simulated at a low-level are somewhat detached from the experimental section. The careful 3D segmentation and annotation surely should lead to stronger realism, but this is not backed up by an experiment.

**Questions:**

My primary question is whether the navigation experiments/techniques are the best way to demonstrate the effectiveness of the overall framework. While they do demonstrate improvements over reasonable VLM-based navigation baselines, I believe that since the main contribution of the work is the novel simulation system which composes "World" and "Agent" layers, perhaps a better way to demonstrate its effectiveness is to show that the system yields data that's more useful in some way, rather than showing improved prompting on VLMs leads to better navigation within the simulation.

Regarding the navigation experiments themselves: more details on the VLM used for the planner, the actual changes of the prompts / agent harness which uses the "Iterative Reasoning". In particular, given the potential difference in base VLM model across the proposed method and baselines, it is important to clarify how much of the difference is coming from the base model.

Minor nit: "while its CR is slightly lower than that of NaVid" --> "while its CR is slightly worse than that of NaVid" (CR is actually higher but therefore worse.)

---

### Official Review · Reviewer_VAbg · 2025-11-02

**Soundness:** 2
**Presentation:** 3
**Contribution:** 1
**Rating:** 2
**Confidence:** 4

**Summary:**

This paper proposes Visually-grounded Humanoid Agents, a two-layer framework combining a semantic 3D world reconstruction layer and a humanoid agent layer for autonomous perception–reasoning–action loops. The method claims to be training-free, relying on off-the-shelf vision-language and motion generation models. Experiments are conducted mainly on navigation and interaction tasks within reconstructed 3D environments.

**Strengths:**

- The idea of connecting world-level 3D reconstruction with humanoid action control is conceptually appealing.

- The modular, training-free design makes the system accessible and easy to reproduce.

**Weaknesses:**

- Ablation and model choice are insufficient.
The system is composed of multiple critical components, yet no detailed ablation is provided. For example, the VLM choice is fixed to Qwen2.5-VL, with no comparison to other open-source (e.g., LLaVA, InternVL) or closed-source (e.g., GPT-4V, Gemini) models. Without this, it’s unclear whether performance comes from the system design or the specific backbone.

- Visual prompting is not novel.
The idea of spatial visual prompting has already been introduced in prior work, notably PIVOT: Iterative Visual Prompting Elicits Actionable Knowledge for VLMs and related methods. The only claimed difference here is extending it to 3D, but there is no experiment showing the effect of “with vs. without 3D.”

- Task and dataset are too narrow.
Evaluation is limited to a self-built navigation benchmark. It is unclear how well the proposed system generalizes. The authors should have compared on broader embodied navigation datasets such as NaVILA-Bench, or other standard 3D navigation tasks.

The novelty and contribution are weak. While the system integrates existing components in a coherent pipeline, there is little methodological innovation beyond prior spatial prompting and training-free composition ideas. The experimental validation is narrow and insufficient to support the claimed story.

**Questions:**

See Weaknesses

---

### Note · Authors · 2025-11-12

I have read and agree with the venue's withdrawal policy on behalf of myself and my co-authors.